# Integration of Proteomic and Metabolomic Data Reveals the Lipid Metabolism Disorder in the Liver of Rats Exposed to Simulated Microgravity

**DOI:** 10.3390/biom14060682

**Published:** 2024-06-12

**Authors:** Mengyao Ru, Jun He, Yungang Bai, Kun Zhang, Qianqian Shi, Fang Gao, Yunying Wang, Baoli Li, Lan Shen

**Affiliations:** 1School of Basic Medicine, Yan’an University, Yan’an 716000, China; rumengyao2022@163.com (M.R.); 18966597676@163.com (K.Z.); 2The State Key Laboratory of Cancer Biology, Department of Biochemistry and Molecular Biology, The Fourth Military Medical University, Xi’an 710032, China; sqq15004194219@126.com; 3Department of Anesthesiology, Xi’an No.3 Hospital, The Affiliated Hospital of Northwest University, Xi’an 710018, China; doctorhejun@163.com; 4Department of Aerospace Medicine, The Fourth Military Medical University, Xi’an 710032, China; baiyun_1123@163.com (Y.B.); yunyingwang@fmmu.edu.cn (Y.W.); 5School of Life Sciences, Yan’an University, Yan’an 716000, China; 6Department of Neurobiology, The Fourth Military Medical University, Xi’an 710032, China; fanggao@fmmu.edu.cn; 7Yan’an Key Laboratory of Microbial Drug Innovation and Transformation, Yan’an University, Yan’an 716000, China

**Keywords:** simulated microgravity, proteomics, metabolomics, lipid metabolism

## Abstract

Long-term exposure to microgravity is considered to cause liver lipid accumulation, thereby increasing the risk of non-alcoholic fatty liver disease (NAFLD) among astronauts. However, the reasons for this persistence of symptoms remain insufficiently investigated. In this study, we used tandem mass tag (TMT)-based quantitative proteomics techniques, as well as non-targeted metabolomics techniques based on liquid chromatography–tandem mass spectrometry (LC–MS/MS), to comprehensively analyse the relative expression levels of proteins and the abundance of metabolites associated with lipid accumulation in rat liver tissues under simulated microgravity conditions. The differential analysis revealed 63 proteins and 150 metabolites between the simulated microgravity group and the control group. By integrating differentially expressed proteins and metabolites and performing pathway enrichment analysis, we revealed the dysregulation of major metabolic pathways under simulated microgravity conditions, including the biosynthesis of unsaturated fatty acids, linoleic acid metabolism, steroid hormone biosynthesis and butanoate metabolism, indicating disrupted liver metabolism in rats due to weightlessness. Finally, we examined differentially expressed proteins associated with lipid metabolism in the liver of rats exposed to stimulated microgravity. These findings contribute to identifying the key molecules affected by microgravity and could guide the design of rational nutritional or pharmacological countermeasures for astronauts.

## 1. Introduction

Space exploration has presented novel opportunities and challenges to humankind. The adverse conditions experienced during space flight, including microgravity, space radiation, extreme temperatures and disruptions in circadian rhythms, pose significant health risks for astronauts [1]. Notably, the alteration of gravity not only affects the body fluid distribution but also leads to bone loss [2,3], muscle atrophy [4], metabolic disorders [5], immune dysfunction [6], changes in haemodynamics [7] and other changes. The liver is a highly metabolically active organ involved in crucial biological processes, such as glucose metabolism, lipid metabolism and biotransformation. Data collected from actual space missions and the establishment of ground-based simulated microgravity models using laboratory animals have revealed the substantial impacts of microgravity on both human and rodent livers [8], including liver damage, metabolic dysregulation and alterations in biotransformation. Abnormal lipid accumulation in the liver can increase susceptibility to non-alcoholic fatty liver disease (NAFLD) among astronauts [9].

The rapid advancement of omics technology has allowed scientists to explore the risks of spaceflight. Numerous single omics studies have explored the effects of microgravity on the liver. For instance, a study using DNA microarray technology to analyse the genome-wide gene expression profile of HepG2 cells cultured under simulated microgravity conditions revealed that weight loss could cause significant changes in the mRNA expression levels of 139 genes, including the APOA1, APOA2 and APOB genes involved in lipid transport, which were significantly downregulated [10]. A metabolomic analysis of the livers of rhesus macaques subjected to 42 days of head-down bed rest revealed hepatic lipid metabolism disorders [11].

However, due to the substantial changes in mRNA, protein and metabolite expression levels, the combination of multiple omics technologies is crucial to comprehensively analyse potential biological relationships and propose effective intervention strategies essential for astronaut health [12,13]. By combining transcriptomics and metabolomics, researchers investigated alterations in hepatic lipid metabolism in mice following short-term spaceflight and found that the expression levels of genes related to triglyceride biosynthesis and hepatic steatosis increased. Moreover, corresponding changes at both the mRNA and metabolite levels indicated that long-term exposure to the space environment may lead to liver injury and activation of the PPARα-mediated signalling pathway, thereby increasing the risk of non-alcoholic fatty liver disease [14].

Nevertheless, few studies have integrated proteomics with metabolomics technologies to investigate metabolic changes in the livers of rats subjected to simulated microgravity. Proteomics is the study of the overall cellular protein composition and dynamics and can provide information such as the protein structures, functions, interactions, expression levels and posttranslational modifications under specific conditions [15]. Metabolomics aims to investigate the dynamic changes in metabolites in living systems following genetic modification or physiological stimulation [16]. It is an extension of transcriptomics and proteomics and can offer a more direct and accurate reflection of organismal physiological states. The integration of proteomics and metabolomics mutually verifies and complements each other, ultimately identifying key proteins or metabolites involved in crucial metabolic pathways that illuminate the molecular regulatory mechanisms underlying phenotypic changes.

Therefore, as methods to gain a deeper understanding of the effects of microgravity on liver function, we applied a mature simulated weightlessness model in this study, adopted a quantitative proteomics analysis method based on TMT and nontargeted metabolomics technology based on LC–MS/MS, screened differentially expressed proteins and metabolites through a series of bioinformatics analyses, identified potential drug targets and provided profound insights into the study of hepatic metabolic disorders in spaceflight.

## 2. Materials and Methods

### 2.1. Animals and Sample Collection

Male Sprague Dawley rats aged 2 months (200 ± 50 g) were purchased from the Experimental Animal Centre of The Fourth Military Medical University (Xi’an, China) and housed in a room controlled for temperature (22 ± 1 °C), humidity (60% ± 10%) and light/dark cycle (12 h/12 h). The animals were randomly divided into a control group and a tail suspension group and fed for 4 weeks with free access to food and water (6 animals in each group). The animals were left to acclimate to the laboratory environment for 3 days prior to the study.

We used a standard rodent tail suspension model, which was described previously [17,18]. Briefly, the rat’s tail was fixed, washed and dried before being covered with benzoin tincture and rosin three times successively. After air drying, the tail was wrapped in medical tape and bandages, and connected to the pulley on the top of the cage with paper clips. The height of the rat was adjusted so that only its forelimbs were able to touch the cage floor while its hindlimbs were slightly off the floor and the body was elevated at a 30-degree angle to the ground, which allowed the rat to move and obtain food and water freely. All procedures involving animals were approved by the Animal Ethics and Experimental Safety Committee of The Fourth Military Medical University (20220563).

After the four-week tail suspension, the rats were anaesthetised with isoflurane and weighed. Then, whole blood samples (2–3 mL) were extracted from the heart by puncture and placed in anticoagulant tubes containing ethylenediamine tetraacetic acid (EDTA). The supernatant containing plasma was collected after centrifugation at 2000× *g* and 4 °C for 10 min. Livers were removed rapidly after rats were perfused with PBS, snap frozen in liquid nitrogen and stored at −80 °C for biochemical studies. The concentrations of triglycerides (TGs), cholesterol (CHOL), low-density lipoprotein (LDL), total protein (TP), albumin (ALB), total bile acid (TBA) and high-density lipoprotein (HDL), as well as the liver injury markers glutamic oxaloyl aminotransferase (AST) and glutamic acid aminotransferase (ALT), were measured with an automatic biochemical analyser (Beckman Coulter, Brea, CA, USA). The livers from 3 rats in each group were used for proteomics analysis and Western blot analysis, and the livers from 6 rats in each group were used for metabolomics analysis.

### 2.2. Oil Red O Staining

Rat liver tissues were prepared with OCT embedding agent before being cut into frozen sections and then stained with Oil Red O (ORO). Whole-slide imaging was performed using a Pannoramic scanner from 3D Histech, and the ORO-positive regions (red) of each slide at the same magnification were quantified by colour deconvolution quantification using Image J (V.1.8.0).

### 2.3. Protein Extraction, Digestion and Tandem Mass Tag (TMT) Labelling

Rat liver tissues were ground individually in liquid nitrogen and lysed with PASP lysis buffer (100 mM NH_4_HCO_3_, 8 M urea, pH 8), followed by 5 min of ultrasonication on ice. After centrifuging the lysate at 12,000× *g* for 15 min at 4 °C, the supernatant was reduced with 10 mM Dithiothreitol (#D9163-25G, Sigma-Aldrich, St. Louis, MO, USA) for 1 h at 56 °C, and subsequently alkylated with IAM (#I6125-25G, Sigma-Aldrich, St. Louis, MO, USA) for 1 h at room temperature in the dark. Then, the precooled acetone equivalent to four times the volume of tissue was added and completely mixed by vortexing. After incubating at −20 °C for 2 h, the mixture was centrifuged at 12,000× *g* at 4 °C for 15 min and the precipitation was collected. Finally, it was washed with 1 mL cold acetone and the particles were dissolved in the solution buffer (8 M urea, 100 mM TEAB, pH 8.5) [19,20,21]. The protein concentration was determined with a BCA kit (Beyotime, Nanjing, China).

Each protein sample was gathered and the volume was adjusted to 100 μL with dissolution buffer. Trypsin and 100 mM TEAB buffer were added for digestion at 37 °C for 4 h, and then trypsin and CaCl_2_ were added to continue the digestion process overnight. After adjusting the pH of the digested sample to 3.0 with formic acid (FA) and centrifuging at 12,000× *g* for 5 min, the supernatant was slowly added to the C18 desalting column, washed with washing buffer (0.1% FA, 3% acetonitrile) 3 times and then eluted with elution buffer (0.1% FA, 70% acetonitrile). The eluents of each sample were collected and lyophilised. Subsequently, the sample was reconstructed with 100 mM TEAB buffer 100 μL and tagged with TMT labelled reagent (Thermo Fisher Scientific, Waltham, MA, USA) dissolved in acetonitrile 41 μL. After shaking for 2 h at room temperature, 8% ammonia was added to stop the reaction. Lastly, all labelled samples were mixed in equal volumes, desalted and lyophilised [22].

### 2.4. Separation of Fractions and LC–MS/MS Analysis

The sample was fractionated by gradient elution using a C18 column (Waters BEH C18, 4.6 × 250 mm, 5 μm) on Rigol L3000 HPLC system with mobile phases A (2% acetonitrile, adjusted pH to 10.0 with ammonium hydroxide) and B (98% acetonitrile). The column oven temperature was set at 45 °C and the analysis was performed with the following elution gradient: 0–10 min, 3–5% B; 10–30 min, 5–20% B; 30–48 min, 20–40% B; 48–50 min, 40–50% B; 50–53 min, 50–70% B; and 53–54 min, 70–100% B. All the fractions were dissolved in 0.1% (*v*/*v*) FA after vacuum drying.

Proteomics analyses were performed using an EASY-nLC 1200 UHPLC system (Thermo Fisher, USA) coupled with a Q Exactive HF-X mass spectrometer (Thermo Fisher, Waltham, MA, USA) operating in data-dependent acquisition (DDA) mode. With 0.1% FA (Solvent A) and 80% ACN/0.1% FA (Solvent B) as the mobile phases, the peptides were separated at a rate of 600 nL/min according to the following procedure: 6–15% B for 2 min, 15–40% B for 46 min, 40–50% B for 2 min, 50–55% B for 1 min, and 55–100% B for 9 min. The mass spectrum parameters were as follows: a Nanospray Flex ion source with an ion transport capillary temperature of 320 °C and a spray voltage of 2.1 kV. The full scan ranged from *m*/*z* 350 to 1500 with a resolution of 60,000 (at *m*/*z* 200), the automatic gain control (AGC) target value was 3 × 10^6^ and the maximum ion injection time was 20 ms. The 40 precursors with the highest abundance in the full scan were dissociated and fragmented by high-energy collisions and analysed via MS/MS. The AGC target value was 5 × 10^4^, the maximum ion injection time was 54 ms, the normalised collision energy was set to 32%, the intensity threshold was 1.2 × 10^5^ and the dynamic exclusion parameter was 20 s.

### 2.5. Data Processing, Identification and Functional Analysis of Proteins

The resulting spectra from each run were searched separately against the UniProt *Rattus norvegicus* (36268 sequences, 2022.1) database with Proteome Discoverer (PD) 2.4 (Thermo Fisher, Waltham, MA, USA). The search parameters were set as follows: the mass tolerance was 10 ppm for precursor ions and 0.02 Da for product ions. Carbamidomethylation was fixed modification, the oxidation of methionine (M) and TMT plex were set to variable modifications and Acetylation, TMT plex, Met loss and Met loss+Acetyl were considered as N-terminal modifications in PD 2.4. A miss of up to 2 cleavage sites was tolerated. The identification and quantification of proteins were carried out by PD 2.4 with a 1% peptide false discovery rate (FDR). The protein quantitation results were statistically analysed using Student’s t test. Proteins whose levels were significantly different between the experimental and control groups (*p* ≤ 0.05 and fold change (FC) ≥ 1.1 or ≤0.91) were defined as differentially expressed proteins.

Gene Ontology (GO) functional analysis was conducted using the InterproScan program (version 5) by searching against the nonredundant protein databases, and the Kyoto Encyclopedia of Genes and Genomes (KEGG) database was used to analyse the protein families and pathways [23]. The differentially expressed proteins were analysed by constructing a volcano map and cluster heatmap and performing GO and KEGG enrichment analyses [24].

### 2.6. Metabolite Extraction and UHPLC–MS/MS Analysis

Metabolites from rat liver tissue were extracted using the method described by Want, E.J. et al. [25]. In short, 100 mg of rat liver tissues were ground with liquid nitrogen individually and then resuspended in precooled 80% methanol by vortexing. The samples were incubated on ice for 5 min and then centrifuged at 15,000× *g* and 4 °C for 20 min. Part of the supernatant was diluted to a final concentration containing 53% methanol with LC–MS-grade water, transferred to a fresh Eppendorf tube and lastly centrifuged at 15,000× *g* and 4 °C for 20 min for analysis in the LC–MS/MS system. An equal volume of liquid from each sample was mixed as a QC sample, and a 53% aqueous methanol solution was added in the same manner as a blank sample.

The metabolomic analysis was performed using a Vanquish UHPLC system (Thermo Fisher, Waltham, MA, USA) coupled to an Orbitrap Q Exactive HF-X mass spectrometer (Thermo Fisher, Waltham, MA, USA) in both positive and negative ion modes. The eluents were eluent A (0.1% FA in water) and eluent B (methanol) for the positive polarity mode, and eluent A (5 mM ammonium acetate, pH 9.0) and eluent B (methanol) for the negative polarity mode. The samples were eluted at a flow rate of 0.2 mL/min with a linear gradient of 17 min according to the following solvent gradient: 2% B, 1.5 min; 2–85% B, 3 min; 85–100% B, 10 min; 100–2% B, 10.1 min; and 2% B, 12 min. The mass spectrum parameters were as follows: a spray voltage of 3.5 kV, capillary temperature of 320 °C, sheath gas flow rate of 35 psi, aux gas flow rate of 10 L/min, S-lens RF level of 60 and aux gas heater temperature of 350 °C.

### 2.7. Data Processing, Identification and Functional Analysis of Metabolites

Compound Discoverer 3.1 (Thermo Fisher, Waltham, MA, USA) was used to analyse the raw data files, including peak alignment, peak picking, normalisation and quantification for each metabolite. Peak intensities normalised to total spectral intensity were matched with mzCloud, mzVault and MassList databases to obtain accurate qualitative and relative quantitative results. The main parameters were set as follows: retention time tolerance, 0.2 min; actual mass tolerance, 5 ppm; signal intensity tolerance, 30%; signal/noise ratio, 3. Statistical analyses were performed using the statistical software R (R version R-3.4.3), Python (Python 2.7.6 version) and CentOS (CentOS release 6.6).

According to the partial least squares discriminant analysis (PLS-DA) model, variable importance in projection (VIP) values were obtained to identify the differentially abundant metabolites between the experimental group and the control group. Furthermore, 200 permutation tests were performed to assess whether the model was overfitting. In addition, we calculated the P value and the FC value, and the metabolites with VIP > 1, *p* < 0.05 and FC ≥ 2 or ≤0.5 were considered differentially abundant metabolites. These metabolites were annotated using the KEGG, HMDB and LIPIDMaps databases.

### 2.8. Combined Proteomic and Metabolomic Analyses

Data standardisation is necessary to eliminate the impact of orders of magnitude and ensure sample consistency. Therefore, three replicates for metabolomics were selected based on the PCA results for a correlation analysis with proteomics. The Pearson correlation coefficient was calculated to analyse the correlation between differentially expressed proteins and differentially abundant metabolites, and the correlation coefficient (R^2^) and *p* value were calculated. R^2^ > 0 was considered to indicate a positive correlation, and R^2^ < 0 was considered to indicate a negative correlation. Based on the results of the KEGG enrichment analysis of the differentially expressed proteins and metabolites, the coenriched pathways were identified and visualised using iPath (Interactive Pathways Explorer version 3) (https://pathways.embl.de/, accessed on 18 April 2024).

### 2.9. Protein Extraction and Western Blotting

Total protein was extracted from rat liver tissues with RIPA buffer (Beyotime Biotechnology, Nantong, China) supplemented with a protease inhibitor cocktail (Roche, Basel, Switzerland), and a BCA assay (EpiZyme, Shanghai, China) was used to determine the total protein content. Proteins were separated by 7.5% sodium dodecyl sulfate–polyacrylamide gel electrophoresis and then transferred to nitrocellulose filter membranes. The membranes were initially blocked with 7% non-fat milk (Boster Biological Technology, Wuhan, China) at room temperature for 1 h; incubated with primary antibodies against Acot2 (1:800, #38596A14, Invitrogen, Waltham, MA, USA), Cpt2 (1:1000, #26555-1-AP, Proteintech, Wuhan, China), and β-actin (1:1000, #20536-1-AP, Proteintech, Wuhan, China) overnight at 4 °C; and then washed with TBST 3 times for 10 min each. Subsequently, the membranes were incubated with HRP-conjugated goat anti-rabbit IgG secondary antibodies at room temperature for 1 h. After 3 washes with TBST, the blots were visualised with ECL detection reagents (NCM Biotech, Beijing, China), and images were acquired using a chemiluminescence imaging system (Tanon 5500, version 1.0). After a grey scale analysis using Image J (V1.8.0), the band density was normalised to that of the internal reference β-actin.

### 2.10. Statistical and Bioinformatic Analyses

Student’s t test was used for two-group comparisons of all non-omics data. Statistical analyses were completed using the statistical software package GraphPad (version 8.1). Principal component analysis (PCA) and PLS-DA were performed with metaX [26]. Volcano plots, bubble diagrams, cluster heatmaps and metabolite–protein correlation network diagrams were generated in R, among which volcano plots and bubble diagrams were generated with ggplot2, metabolite–protein correlation networks were mapped with mixOmics packages and the heatmap package was used to construct cluster heatmaps. The data are presented as the means ± standard deviations (SDs) [27]. *p* < 0.05 was considered to indicate statistical significance.

## 3. Results

### 3.1. Establishment of a Simulated Microgravity Model and Hepatic Lipid Accumulation

The simulated microgravity model was established using the method described above, and the weight changes of the rats were measured four weeks later. The tail suspension group exhibited significant weight loss (*p* < 0.05, Figure 1A). The loss of muscle mass, the decrease in bone density, reduced food and water intake may contribute to weight loss in rats after the tail suspension test [5,14]. Subsequently, the rat livers were stained with ORO, and the positive staining intensity was quantified with Image J (three fields per section, 14× magnification for image analysis, Figure 1B,C). The results show that the ORO-positive area in the tail suspension group was significantly increased compared with the control group (*p* < 0.01). These findings are consistent with previous studies suggesting that hepatic lipid accumulation may be attributed to microgravity [9,14].

In terms of plasma lipid metabolite levels, TG (*p* = 0.0337), CHOL (*p* = 0.0039) and LDL (*p* = 0.0005) levels were significantly increased in the TS group, while no significant differences in TP, ALB, TBA and HDL levels or the liver injury indices ALT and AST were observed (Figure 2). Therefore, four weeks of tail suspension may not be sufficient to induce significant changes in liver function.

### 3.2. Identification and Functional Enrichment Analysis of Differentially Expressed Proteins

We employed TMT technology to analyse liver tissue samples from rats in the control and tail suspension groups and to investigate the impact of weightlessness on protein levels. A total of 4203 proteins were identified, and the quantitative results are shown in Appendix A. PCA of the proteomic data revealed a significant separation between the experimental and control groups, indicating substantial changes in proteomic data during tail suspension (Figure 3A). Based on the criteria of FC (≥1.1 or ≤0.91) and *p* values obtained using *t* tests (≤0.05), we observed significant differences in 63 proteins, with the levels of 31 proteins increasing and the levels of 32 proteins decreasing (Figure 3B). Complete lists of the differentially expressed proteins are provided in the Appendix A (Appendix A). Furthermore, a cluster analysis was performed based on the relative contents of differentially expressed proteins, resulting in a clustering heatmap, as depicted in Figure 3C. The protein expression profiles were distinctly divided into two categories, suggesting that the liver protein expression patterns of the simulated microgravity group were notably different.

Next, GO functional significance enrichment analysis was conducted to annotate these differentially expressed proteins (Figure 3D) (Appendix A). The analysis revealed six significantly enriched GO terms for these differentially expressed proteins in the biological process category, 3 in the cellular component category and 12 in the molecular function category. A series of lipid metabolism-related processes, such as cellular lipid biosynthesis, phosphatidylserine biosynthesis, 3-hydroxy-3-methylglutaryl-CoA reductase (Hmgcr) activity and 3-hydroxy-3-methylglutaryl-CoA synthase (Hmgcs) activity, were altered. Notably, several differentially expressed proteins, including acyl-CoA thioesterase 2 (Acot2), carnitine palmitoyl transferase 2 (Cpt2), acetyl-CoA acyltransferase 1b (Acaa1b), 3-hydroxy-3-methylglutaryl-CoA lyase (Hmgcl), phosphatidylserine synthase 1 (Ptdss1), Hmgcr and Hmgcs2, were shown to be involved in the above processes.

KEGG pathway enrichment analysis was used to identify major biochemical metabolic pathways and signal transduction pathways associated with these significantly altered proteins (Figure 3E). The three most significant KEGG pathways were the ketone synthesis and degradation pathway, the alpha-linolenic acid metabolism pathway and the terpenoid backbone biosynthesis pathway. The enrichment of biosynthesis of unsaturated fatty acids, fatty acid metabolism and degradation and PPAR signalling pathways also revealed the themes of lipid metabolism disorders.

In summary, extensive and comprehensive proteomic data indicate that a simulated microgravity environment can induce significant alterations in the levels of proteins, particularly enzymes involved in lipid metabolism, in rats.

### 3.3. Screening and Functional Enrichment Analysis of Differentially Abundant Metabolites

Through a nontargeted LC–MS/MS metabolomics analysis, a total of 1017 metabolites were detected in the liver tissues from the experimental and control rats (Appendix A). Initially, PCA and PLS-DA were employed to assess sample correlations and differences (Figure 4A,B). Although the PCA scores showed partial overlap within the 95% confidence interval between the simulated microgravity group and the control group, the PLS-DA model revealed a significant separation between the two groups (R^2^ = 0.96, Q^2^ = 0.34), indicating the strong explanatory power and predictive ability of the data. Additionally, the arrangement test results indicate that overfitting was not observed in the data (R^2^ = 0.93, Q^2^ = −0.71, Figure 4C), confirming the good predictability of the PLS-DA model. Variable importance in projection (VIP), representing the projection importance of the first principal component in the PLS-DA model, was used as a criterion for screening differentially expressed metabolites. Based on the criteria of VIP > 1.0, FC > 1.2 or FC < 0.83 and *p* < 0.05, we identified 150 significantly differentially abundant metabolites; among them, 83 were upregulated, while 67 were downregulated (Figure 4D) (Appendix A). The stratified clustering heatmaps revealed substantial variations in metabolite abundance between the experimental group and the control group. Furthermore, increases in the contents of unsaturated fatty acids and glycerophospholipids such as PC, LPG, LPS and other lipid derivatives, including prostaglandin, were detected; moreover, decreases in the levels of metabolites related to lipid metabolism, such as acetic acid and 2-hydroxyisobutyric acid, were detected (Figure 4E).

Based on these findings, we conducted systematic pathway annotations for all identified metabolites (Appendix A) and KEGG enrichment analysis of differentially abundant metabolites to obtain pathway information for these differentially regulated molecules and analyse their functions accordingly. The results of the KEGG analysis indicate that differentially abundant metabolites were mainly enriched in linoleic acid metabolism, steroidogenesis, butanoate metabolism, amino acid metabolism (arginine, proline, histidine, alanine, aspartate and glutamate metabolism), nitrogen metabolism and oxidative phosphorylation (Figure 4F).

### 3.4. Combined Proteomic and Metabolomic Analyses

The correlation analysis of the differentially expressed proteins and metabolites showed that dehydrocholic acid and 11-deoxyprostaglandin F1α were significantly positively correlated with Acot2, Hmgcl, Hmgcs2, Cpt2 and Acaa1b (*p* < 0.05). Fluprostenol serinol amide was significantly negatively correlated with Hmgcl and Acaa1b (*p* < 0.05) but significantly positively correlated with Acox3 and Ptdss1 (*p* < 0.05) (Figure 5A–C). KEGG pathway analysis of the differentially expressed proteins and metabolites indicated that the primary biochemical pathways they jointly participated in were the biosynthesis of unsaturated fatty acids, linoleic acid metabolism, steroid hormone biosynthesis and butanoate metabolism, as shown in Figure 5D. The coenriched pathways were visualised via iPath and are shown in Appendix A.

### 3.5. Experimental Verification of Differentially Expressed Proteins

Moreover, we further verified the expression of key lipid metabolism enzymes (Acot2 and Cpt2) in the control and tail-suspended groups by performing Western blotting. β-Actin was selected as the internal reference. Consistent with the proteomic data, the protein expression levels of Acot2 and Cpt2 in the tail-suspended group were significantly decreased compared with those in the control group (*p* < 0.05, Figure 6A,B).

## 4. Discussion

In this study, we used TMT-labelled quantitative proteomics technology and LC–MS/MS-based metabolomics technology to comprehensively analyse and identify the differentially expressed proteins and metabolites in the livers of rats exposed to a simulated microgravity environment and then analysed the biological functions of these differentially expressed proteins and metabolites via bioinformatics. The aim of this study was to identify key regulatory molecules and signalling pathways involved in abnormal liver metabolism induced by a microgravity environment.

The livers of mice on the Space Shuttle mission (STS-135, 13 days) and the International Space Station mission (RR-1 NASA, 37 days; RR-1 CASIS, 21 days) showed higher ORO staining than ground controls, indicating lipid accumulation in the liver. Furthermore, a two-way ANOVA with mission (i.e., RR-1 NASA, RR-1 CASIS, RR-3) and flight conditions (i.e., flight versus ground) as two distinct factors revealed that the microgravity environment is the most primary factor influencing the level of hepatic lipid accumulation [9,14]. Our findings were consistent with these results, which further confirmed that lipids accumulated in the liver of a simulated weightlessness rat model. PPARs are ligand-activated transcription factors belonging to the nuclear hormone receptor family. Upon binding to their ligands, these receptors translocate into the nucleus where they form heterodimers with the retinol X receptor (RXR), subsequently binding to peroxisome proliferating response elements (PPREs) in target gene promoters to regulate gene transcription and function [28,29,30]. PPARs regulate the expression of many target genes involved in lipid metabolism, such as genes involved in fatty acid uptake and transport (Fab and CD36), fatty acid oxidation of peroxisomes and mitochondria (Acox and Cpt), breakdown of triglyceride-rich lipoproteins (LPL and Apo) and ketogenesis under starvation or fasting conditions (Hmgcs and Hmgcl) [31,32,33]. In this study, we observed decreases in the abundances of Acaa1b, Acot2 and Cpt2 and an increase in the abundance of Ptdss1. Acaa1b, Acot2 and Cpt2 are downstream target genes regulated by PPARs, and the upregulation of the expression of these genes could increase the rate of fatty acid β-oxidation and alleviate abnormal lipid accumulation. Ptdss1 primarily participates in glycerophospholipid metabolism, which synthesises phosphatidylserine (PS) via the exchange of L-serine for the choline head group of phosphatidylcholine (PC) [34]. Therefore, the increase in Ptdss1 expression may be responsible for the alterations in glycerophospholipid metabolism. Furthermore, GO and KEGG enrichment analyses revealed the significant enrichment of terms related to phosphatidylserine biosynthetic processes, unsaturated fatty acid biosynthesis, fatty acid metabolism and the PPAR signalling pathway. The supporting metabolomics data revealed increased levels of lipid metabolites, including unsaturated fatty acids (docosapentaenoic acid and adrenic acid), glycerophospholipids (PC, LPS, LPG and LPC) and prostaglandins, which result in dysregulated lipid metabolism. These findings highlight the perturbation of lipid metabolism, particularly that of fatty acids and glycerophospholipids.

Moreover, the reduction in the fatty acid oxidation rate affects the formation of ketone bodies and acetyl-CoA. Notably, this study also revealed significant enrichment in the production and degradation of ketone bodies. Ketone bodies serve as crucial energy substrates for the brain, while the liver is the exclusive organ involved in ketone generation. An insufficient supply of ketone bodies has been implicated in neurodegenerative diseases such as Alzheimer’s disease [35]. At the protein level, we observed decreased contents of Hmgcs and Hmgcl, key enzymes involved in the production of ketone bodies. Interestingly, the cluster heatmap of differentially abundant metabolites revealed reduced levels of methyl-2-hydroxyisobutyric acid, which structurally resembles 3-hydroxybutyric acid (3HB), a major component of ketone bodies. Collectively, these results indicate that prolonged exposure to microgravity weakened hepatic lipid catabolic metabolism, which led to an inadequate availability of ketone bodies. In addition, 3HB and its hydroxybutyrate methyl ester (3-hydroxybutyrate methyl ester, 3HBME) inhibit preosteoclast differentiation by inhibiting the activation of nuclear factor of activated T cells cytoplasmic 1 (NFATc1) under simulated microgravity conditions. Additionally, preventing osteoclastic bone resorption, reducing calcium loss from bones and promoting the growth and differentiation of MC3T3-E1 osteoblasts in vitro can ameliorate ovariectomy-induced osteoporosis in rats [36,37]. Therefore, a reduction in ketone bodies may exacerbate bone loss in astronauts during spaceflight missions or upon re-entry to Earth.

In fact, scientists have provided copious evidence that microgravity induces hepatic lipid accumulation by actual spaceflight or by establishing simulated microgravity models on the ground. A transcriptomic and proteomic analysis which utilised mice liver space flight samples demonstrated that NAFLD-related pathways were activated and resulted in hepatic lipid accumulation [9]. They considered that the liver is the origin of lipid dysregulation in space flight, and that major perturbations in key regulatory genes regulating lipid and fatty acid metabolism are frequently regarded as the “first-hit” of NAFLD, which is characterised by insulin resistance and disruption of fatty acid metabolism. Downregulation of the PPARα signalling pathway found in transcriptomic data is associated with the “second hit” phase of NAFLD. The phase is characterised by inflammation and fibrination, which is also involved in mitochondrial dysfunction and oxidative stress. Data from the NASA Twin Study and astronaut research also highlight mitochondrial dysregulation as a central hub for space biology [38,39]. They strongly support the hypothesis that mitochondrial stress caused by space flight may impact lipid metabolism processes in mice and humans.

For astronauts, countermeasures to improve this metabolic disorder include exercises and nutritional or therapeutic intervention. Exercise is the principal countermeasure as moderate physical exercise can relieve metabolic stress [40]. In addition to exercise, nutritional interventions and pharmacotherapy would be considered. An essential aspect of the metabolic health of astronauts is to maintain energy balance by developing a rational nutrition strategy to consume the recommended energy intake [5]; moreover, rational use of drugs can effectively prevent or treat lipid accumulation.

However, our study has several limitations. First, due to the limited number of samples, our experimental results may have certain limitations, and the molecular expression levels could be further verified after expanding the study cohort. Second, while our combined analysis of proteomic and metabolomic data provides insights into various biological processes, the use of lipid metabolomics technology is necessary for a comprehensive analysis and detection of changes specifically related to lipid metabolism. Finally, considering the genetic differences and physiological and metabolic variations between rodents and humans, as well as the potential shortcomings associated with tail suspension models, the metabolic changes in rats under the influence of microgravity cannot fully summarise the human response to space flight.

## 5. Conclusions

In this study, we established a ground simulated microgravity model by tail suspension, and we demonstrated that microgravity induces hepatic lipid accumulation. In addition, by combining proteomics and metabolomics, we identified the changes in several important biomolecules in the livers of tail-suspended rats. We integrated the differentially expressed protein and metabolite enrichment analyses. Notably, stimulated microgravity leads to direct and indirect abnormal alterations in proteins and metabolites involved in the PPAR signalling pathway. These changes may be the primary cause of lipid metabolism disorders in rat livers. Overall, these findings contribute to a better understanding of the effects of microgravity on hepatic lipid metabolism, and guiding astronauts to rational nutritional or pharmaceutical interventions.

## Figures and Tables

**Figure 1 biomolecules-14-00682-f001:**
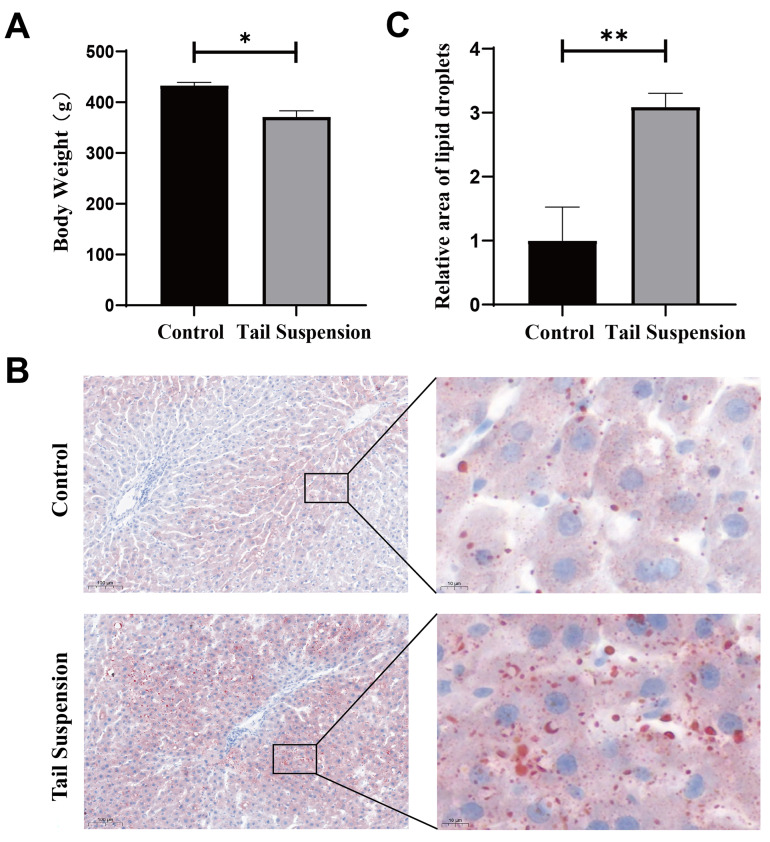
The lipid content increased significantly in the rat liver after the tail suspension test. (**A**) The weight of the rats was measured after four weeks of tail suspension. (**B**) Oil Red O (ORO) staining of the livers of rats in the control and tail-suspended groups (10×, left panel; 80×, right panel). (**C**) Relative quantitative analysis of the ORO staining results (n = 5 fields (3 rats)). The data are presented as the means ± SDs. * *p* < 0.05 and ** *p* < 0.01.

**Figure 2 biomolecules-14-00682-f002:**
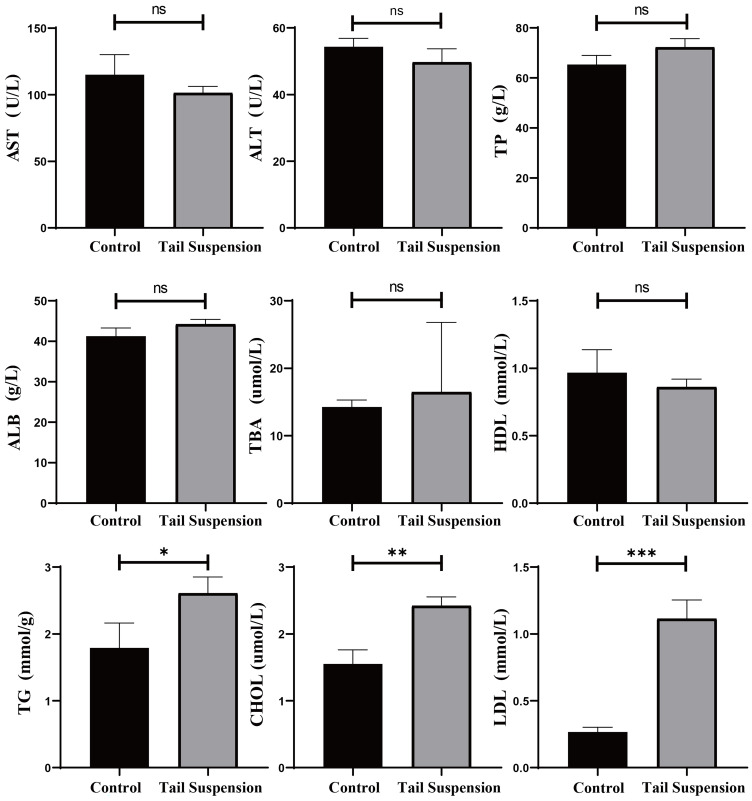
Results of biochemical analyses of the peripheral blood from the rats. The data are presented as the means ± SDs (n = 3 per group). * *p* < 0.05, ** *p* < 0.01, *** *p* < 0.001, and ns not significant.

**Figure 3 biomolecules-14-00682-f003:**
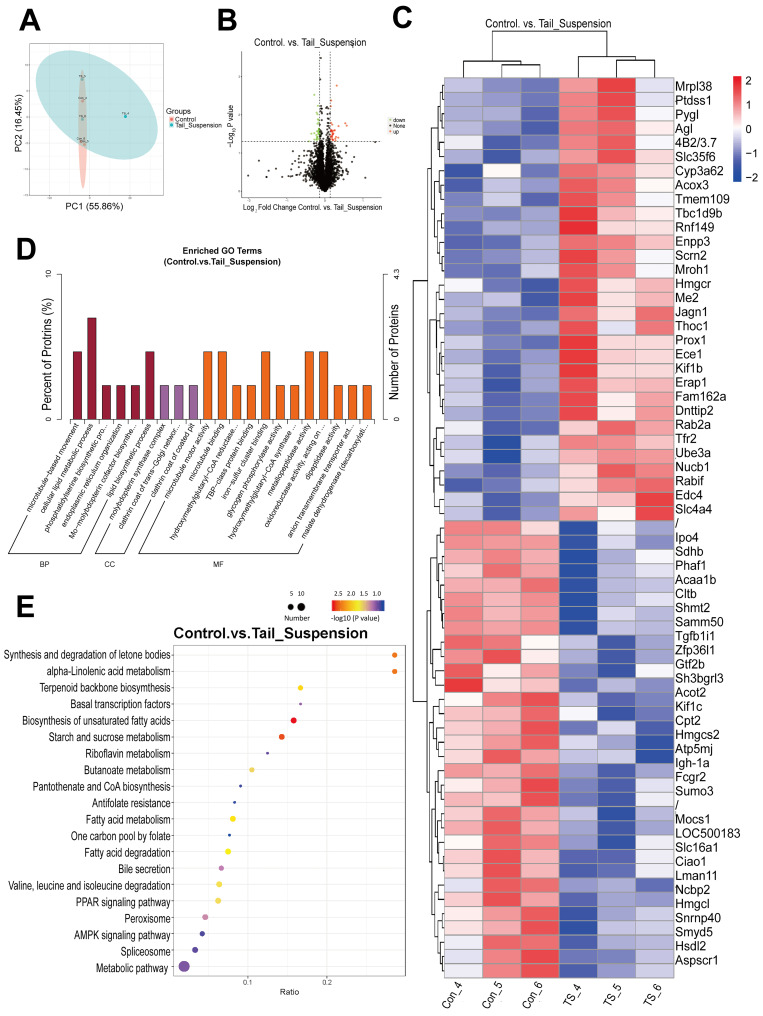
Identification and functional enrichment analysis of differentially expressed proteins in the livers of rats after the tail suspension test. (**A**) Principal component analysis of the protein abundances of proteins quantified in rat liver tissue. The abscissa PC1 and ordinate PC2 represent the scores of the first and second principal components, respectively, and the ellipse is the 95% confidence interval. (**B**) Volcano plots of differentially expressed proteins in the livers of rats in the control and tail-suspended groups. The horizontal axis indicates the fold change (FC) in the differentially expressed proteins (log2 FC), the vertical axis indicates the *p* value (−log10 *p* value), and the black dot represents a protein with no significant difference. T. the red dots represent the upregulated proteins, and the green dots represent the downregulated proteins. (**C**) Cluster heatmaps of differentially expressed proteins in the livers of rats in the control and tail-suspended groups. Z scores are corrected for each line. The vertical axis represents significant differentially expressed proteins, and the horizontal axis represents sample information. (**D**) GO enrichment histogram of differentially expressed proteins in the livers of rats in the control and tail-suspended groups. The enrichment results are shown for three categories: biological process (BP), cell component (CC) and molecular function (MF) (*p* value ≤ 0.05). The percentage of the ordinate represents the number of differentially expressed proteins annotated to a GO term as a percentage of the number of differentially expressed proteins annotated to all proteins with GO annotation information. (**E**) KEGG enrichment bubble diagrams of differentially expressed proteins in the livers of rats in the control and tail-suspended groups. The horizontal axis is the ratio of the number of differentially expressed proteins in the corresponding pathway to the total number of proteins identified in the pathway. The larger the value is, the greater the degree of enrichment of differentially expressed proteins in the pathway. The colour of the points represents the *p* value of the hypergeometric test. The colour ranges from blue to red. The redder the colour is, the smaller the *p* value, indicating greater reliability of the test and greater statistical significance. The size of the dot represents the number of different proteins in the corresponding pathway, and the larger the dot is, the greater the number of differentially expressed proteins in the pathway.

**Figure 4 biomolecules-14-00682-f004:**
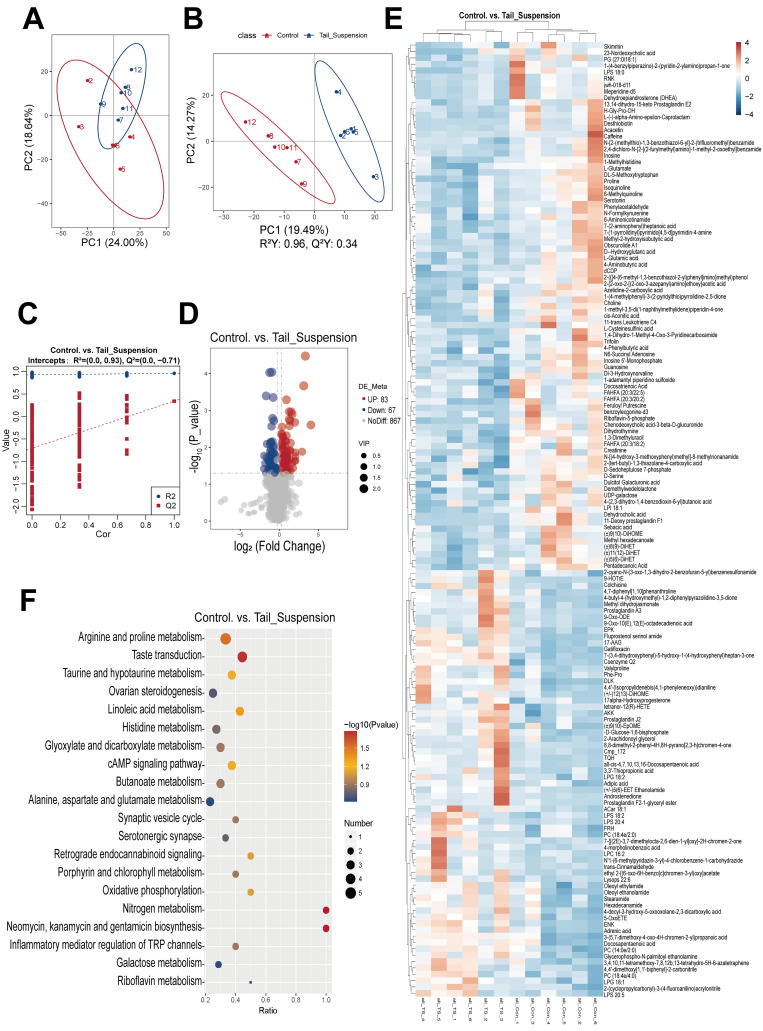
Screening and functional enrichment analysis of differentially abundant metabolites in the livers of rats after the tail suspension test. (**A**) Principal component analysis of metabolites in rat liver tissue. (**B**) Score plots of the PLS-DA model of metabolites. The abscissa is the score of the sample on the first principal component, and the ordinate is the score of the sample on the second principal component. (**C**) Arrangement tests of PLS-DA models. The abscissa represents the correlation between the random group Y and the original group Y, and the ordinate represents the R^2^ and Q^2^ scores. (**D**) Volcano plots of differentially abundant metabolites in the livers of rats in the control and tail-suspended groups. The upregulated metabolites are represented by red dots, and the downregulated metabolites are represented by blue dots. The size of the dot represents the VIP value. (**E**) Cluster heatmaps of differentially abundant metabolites in the livers of rats in the control and tail-suspended groups. (**F**) KEGG enrichment bubble diagrams of differentially abundant metabolites in the livers of rats in the control and tail-suspended groups.

**Figure 5 biomolecules-14-00682-f005:**
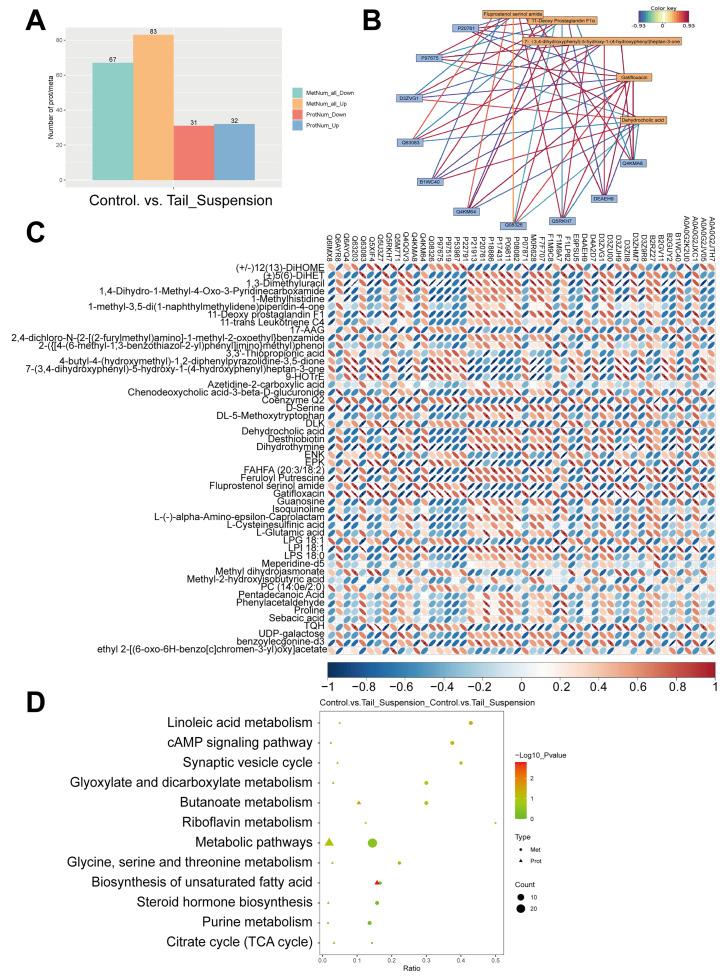
The combination of proteomics and metabolomics data. (**A**) The histogram summarises the number of differentially expressed proteins and metabolites. (**B**) Correlation network diagram of differentially expressed proteins and metabolites. It was constructed using the top 5 enriched metabolites from the metabolomics enrichment analysis and top 10 enriched proteins from the proteomics enrichment analysis. Metabolites are represented by yellow boxes, while proteins are represented by blue boxes; red lines denote positive correlations, and blue lines denote negative correlations. The width and darkness of the line correspond to the magnitude of the correlation coefficient. (**C**) Heatmaps of correlations between differentially expressed proteins and metabolites. The 50 most significantly enriched proteins and metabolites are shown (*p* values ranked from smallest to largest). The vertical axis represents differentially abundant metabolites, while the horizontal axis represents differentially expressed proteins. Blue indicates a negative correlation, and red indicates a positive correlation. The intensity of the colour reflects the strength of the correlation, and flatter ellipses indicate higher absolute values for the correlation. A statistically significant correlation is denoted by * for *p* < 0.05. (**D**) Bubble map of the KEGG enrichment analysis of differentially expressed proteins and metabolites.

**Figure 6 biomolecules-14-00682-f006:**
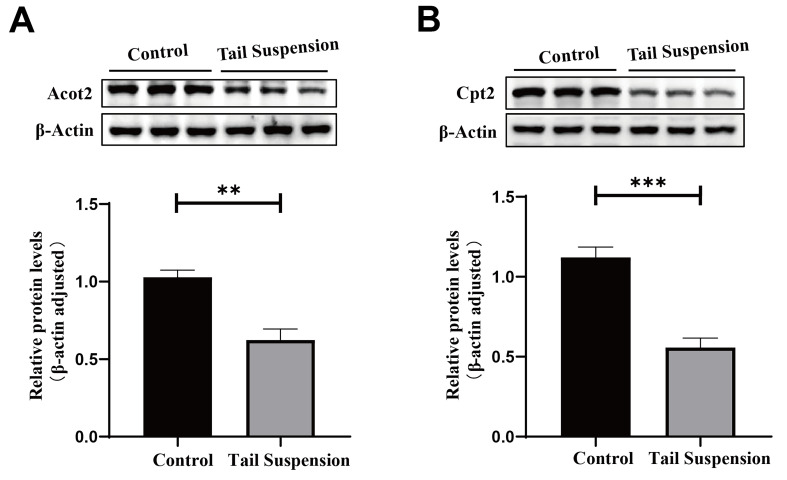
Western blot analysis of molecules whose expression changed significantly in the livers of rats in the control and tail-suspended groups. The expression of acyl-CoA thioesterase 2 (Acot2) (**A**) and carnitine palmitoyl transferase 2 (Cpt2) (**B**) was detected by performing Western blot analysis and quantified. The data are presented as the means ± SDs; ** *p* < 0.01, *** *p* < 0.001.

## Data Availability

All relevant data of this study are given in the manuscript and the Appendix A. Additional data will be provided upon request.

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
