# Peer review of "Integration of Proteomic and Metabolomic Data Reveals the Lipid Metabolism Disorder in the Liver of Rats Exposed to Simulated Microgravity"

_biomolecules, 2024, doi:10.3390/biom14060682_

Round 1

Reviewer 1 Report

Comments and Suggestions for Authors

The authors present a thoroughly planned and conducted multi-omic study comparing liver samples from rats subjected to tail suspension as a simulation of microgravity and rats held under standard conditions.

The experiment design and rationale is well explained and the results are presented in a clear way and are of relevance to the field.

There are, however, three minor points that the authors might want to address:

1.) I would recommend to present the data as mean +- SD and not SEM, as this accounts better for the actual variance of the data (see also Barde MP, Barde PJ. What to use to express the variability of data: Standard deviation or standard error of mean? Perspect Clin Res. 2012 Jul;3(3):113-6. doi: 10.4103/2229-3485.100662. PMID: 23125963; PMCID: PMC3487226)

2.) It would have been interesting to know whether there were differences in the food intake between the two groups.

3.) The labels on Figs. 3-5 are very hard to read

Apart from this, I have no other comments and I recommend the publication of the manuscript after a minor revision.

Reviewer 2 Report

Comments and Suggestions for Authors

Introduction: Authors state that they will identify potential drug targets (p.2, line 86-87). Is it revealed later, which targets have been detected?

 Methods: Please comment in some more detail on the tail suspension model. Is it really suitable to simulate microgravity?

 Results: was loss of bodyweight caused by simulation of microgravity, or just by immobilization and consecutive loss of muscle mass?

 Discussion: Can authors discuss the consequences of their findings in more detail? The fact that (simulated) microgravity causes fatty liver disease has been known before. Is there a way to counteract this problem?

Are there any theories that can explain accumulation of fat in the liver under simulated or real microgravity?

A conclusion should be provided after the discussion

 Figures: The letters of the figure labelling are very small and therefore hard to read.
